

# Unimodular jordanian deformations of integrable superstrings

Stijn J. van Tongeren

Institut für Physik, Humboldt-Universität zu Berlin,
IRIS Gebäude, Zum Grossen Windkanal 6, 12489 Berlin, Germany

svantongeren@physik.hu-berlin.de

## Abstract

We find new homogeneous $r$ matrices containing supercharges, and use them to find new backgrounds of Yang-Baxter deformed superstrings. We obtain these as limits of unimodular inhomogeneous $r$ matrices and associated deformations of $\mathrm{AdS}_2 \times \mathrm{S}^2 \times \mathrm{T}^6$ and $\mathrm{AdS}_5 \times \mathrm{S}^5$. Our $r$ matrices are jordanian, but also unimodular, and lead to solutions of the regular supergravity equations of motion. In general our deformations are equivalent to particular non-abelian T duality transformations. Curiously, one of our backgrounds is also equivalent to one produced by TsT transformations and an S duality transformation.

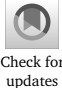
# 1  Introduction

Yang-Baxter sigma models [1,2] are integrable deformations of sigma models, built on $r$ matrices that solve the classical Yang-Baxter equation. There is a variety of $r$ matrices, giving various deformations of integrable sigma models. It is interesting to study these for instance in the case of the AdS$_5$×S$^5$ superstring, where integrability has led to impressive results in testing the AdS/CFT correspondence [3,4].

Yang-Baxter deformations broadly come in two types, namely they can be inhomogeneous [5] of homogeneous [6], leading to trigonometric $q$-deformed [7] or twisted [8] symmetry in the sigma model respectively.[1] Inhomogeneous deformations are unique for compact Lie algebras, but offer some inequivalent options when it comes to noncompact algebras and superalgebras. Homogeneous deformations are quite diverse and offer multiple different options, depending on the algebra under consideration. Classifying the homogeneous solutions of the classical Yang-Baxter equation for a given algebra is an open problem.

The first Yang-Baxter deformation of the AdS$_5$×S$^5$ superstring to be studied was based on the standard inhomogeneous solution of the classical Yang-Baxter equation [5,7,10]. Given the noncompact nature of AdS$_5$×S$^5$ there are two further deformations [7] which appear to differ geometrically while sharing the same integrable structure in terms of e.g. $S$-matrices [11]. Going beyond the bosonic algebra to the full superalgebra, there is some further freedom via the choice of Dynkin diagram, which was recently used to get new deformations with different fermionic sectors [12].

When it comes to homogeneous deformations of the AdS$_5$×S$^5$ superstring, many have been found and studied, see e.g. [13–19]. While inhomogeneous $r$ matrices inherently involve both even and odd generators of a superalgebra, so far the study of homogeneous deformations has been limited to $r$ matrices built out of purely even generators. In this paper we will consider several homogeneous deformations for $r$ matrices involving also odd generators, and determine their effect on the background geometry of the superstring. To find such $r$ matrices and backgrounds we will consider limits of noncompact group transformations, which produce homogeneous $r$ matrices from inhomogeneous ones [20]. This procedure yields $r$ matrices built from even generators when applied to the standard inhomogeneous $r$ matrix, but the results of [12] give new options.

Our results will also answer a standing question regarding the existence of unimodular jordanian $r$ matrices for $\mathfrak{psu}(2,2|4)$. This question arose out of the study of Weyl invariance of Yang-Baxter sigma models [21]. Namely, while Yang-Baxter deformed superstrings have $\kappa$ symmetry [5], the background of the standard inhomogeneous deformed AdS$_5$ × S$^5$ superstring does not satisfy the supergravity equations of motion [22], but rather a generalization thereof [23]. Subsequently it was shown that indeed $\kappa$ symmetry implies only these generalized equations and not the more restrictive standard supergravity equations of motion [24]. This is believed to guarantee scale invariance but not Weyl invariance [23–25], and it was found that in order for a deformed background to solve the more restrictive standard supergravity equations of motion the $r$ matrix needs to satisfy a unimodularity constraint [21,25]. This led to the question whether it is possible to find unimodular extensions of the well known type of jordanian homogeneous $r$ matrices. The question of unimodularity was also the motivation for [12], whose newly found deformations correspond to unimodular $r$ matrices, and hence supergravity backgrounds. Starting from the results of [12] we will find homogeneous $r$ matrices of jordanian type, that are also unimodular.

Unimodularity may or may not be the end of the story regarding Weyl invariance. Recently it was suggested that models whose backgrounds satisfy only the generalized supergravity

---

[1]These types of structures were originally uncovered in particular models, before those models were identified as Yang-Baxter models, see e.g. [9] and references therein.

equations, should nevertheless be Weyl invariant [26], based on earlier results contained in [27, 28]. Subsequently, a local and covariant generalization of the Fradkin-Tseytlin term in terms of world-sheet torsion variables was proposed in [29]. However, as discussed in [29], these proposals have troublesome features, and it is presently not clear whether solutions of the generalized equations truly give Weyl invariant models. In this paper we will remain firmly within the footing of conventional supergravity and Weyl invariance.

We will not give an exhaustive overview of all backgrounds and $r$ matrices that can be obtained from [12] by our procedure. Instead we will focus on several illustrative examples. The first is a simple non-abelian homogeneous unimodular deformation of AdS$_2 \times$S$^2 \times$T$^6$ that turns out to be timelike T dual to undeformed AdS$_2 \times$S$^2 \times$T$^6$ with a nonstandard RR sector. The second is a similar but more involved deformation of AdS$_5 \times$ S$^5$. Finally, we consider a further limit of this deformation of AdS$_5 \times$ S$^5$ which corresponds to a basic unimodular extension to the $r$ matrix considered in [6, 30] and leads precisely to the supergravity background given in [30].

This paper is organized as follows. We start with a short overview of the Yang-Baxter superstring sigma model in section 2, focussing on AdS$_5 \times$S$^5$ for concreteness, and introduce our notation. Next, in section 3 we discuss the basic unimodular extension of a jordanian $r$ matrix. Then, in section 4 we discuss our limiting procedure and use it to extract backgrounds. We finish with further comments and open questions.

## 2 The deformed superstring action

The Yang-Baxter deformation of the AdS$_5 \times$ S$^5$ superstring action takes the form [5][2]

$$S = -\frac{T}{2} \int d\tau d\sigma \tfrac{1}{2} (\sqrt{h} h^{\alpha\beta} - \epsilon^{\alpha\beta}) \text{sTr}(A_\alpha d_+ J_\beta), \tag{1}$$

where $J = (1 - \eta R_g \circ d_+)^{-1}(A)$ with $R_g(X) = g^{-1}R(gXg^{-1})g$. At $\eta = 0$ ($R = 0$) we get the undeformed AdS$_5 \times$ S$^5$ superstring action of [31].

The operator $R$ is a linear map from $\mathfrak{g} = \mathfrak{psu}(2, 2|4)$ to itself. Provided it is antisymmetric,

$$\text{sTr}(R(m)n) = -\text{sTr}(mR(n)), \tag{2}$$

and satisfies the inhomogeneous classical Yang-Baxter equation

$$[R(m), R(n)\} - R([R(m), n\} + [m, R(n)\}) = [m, n\}. \tag{3}$$

The resulting model is classically integrable and has a form of $\kappa$ symmetry [5]. The same is true for solutions of the homogeneous classical Yang-Baxter equation

$$[R(m), R(n)\} - R(R(m), n\} + [m, R(n)\}) = 0, \tag{4}$$

provided we set $\eta$ to zero in $d_\pm$ [6].[3] These equations involve the graded commutator

$$[a, b\} = ab - (-1)^{[a][b]} ba, \tag{5}$$

where $[m]$ denotes the degree of generator $m$, i.e. 0 for even (bosonic) generators and 1 for odd (fermionic) ones.

---

[2]$T$ is the effective string tension, $h$ is the world sheet metric, $\epsilon^{\tau\sigma} = 1$, $A_\alpha = g^{-1}\partial_\alpha g$ with $g \in G = $ PSU(2, 2|4), sTr denotes the supertrace, and $d_\pm = \pm P_1 + \frac{2}{1-\eta^2}P_2 \mp P_3$ where the $P_i$ project onto the $i$th $\mathbb{Z}_4$ graded components of the semi-symmetric space PSU(2, 2|4)/(SO(4, 1) × SO(5)).

[3]This follows directly if we rescale the inhomogeneous $R$ operator as $R = \alpha/(2\eta)\hat{R}$ and consider the limit $\eta \to 0$.

It is convenient to represent $R$ operators by $r$ matrices, mapping the latter to the former via the Killing form of $\mathfrak{g}$ (the supertrace) as

$$R(m) = \mathrm{sTr}_2(r(1 \otimes m)), \tag{6}$$

with

$$r = r^{ij} t_i \wedge t_j \in \mathfrak{g} \otimes \mathfrak{g}, \tag{7}$$

where $\mathrm{sTr}_2$ denotes the supertrace over the second space in the tensor product, the $t_i$ generate $\mathfrak{g}$, and a sum over repeated indices is implied. The wedge denotes a graded antisymmetric tensor product

$$a \wedge b = a \otimes b - (-1)^{[a][b]} b \otimes a. \tag{8}$$

Building $r$ using this graded antisymmetric tensor product is equivalent to the antisymmetry of the $R$ operator of equation (2).

As a map from $\mathfrak{g} = \mathfrak{psu}(2,2|4)$ to itself, $R$ has to satisfy the reality condition

$$(R(x))^\dagger = -HR(x)H, \quad x \in \mathfrak{g}, \tag{9}$$

where $H$ is the metric defining $\mathfrak{g}$, see appendix A. This implies that the coefficients $r^{ij}$ have to satisfy

$$(r^{ij})^* = (-1)^{[t_j]} r^{ij}, \tag{10}$$

since $\mathrm{sTr}(xy)$ is real when $x$ and $y$ are both even, and imaginary otherwise. In other words the $r^{ij}$ are real when the corresponding generators are both even, and the $r^{ij}$ are purely imaginary when the corresponding generators are both odd.[4]

In this notation, the standard solution of inhomogeneous classical Yang-Baxter equation over the complexified algebra $\mathfrak{g}_\mathbb{C}$ is given by

$$r = ie_j \wedge f^j, \tag{11}$$

where the $e_j$ and $f_j$ denote positive and negative roots of $\mathfrak{g}_\mathbb{C}$ respectively. A jordanian solution has the basic structure

$$r = h \wedge e, \quad \text{where } [h,e] = e, \tag{12}$$

while a basic abelian $r$ matrix takes the form

$$r = a \wedge b, \quad \text{where } [a,b] = 0. \tag{13}$$

From the form of the action it is clear that the left global $G$ symmetry of the undeformed model is broken to the group generated by those $t$ for which for all $x \in \mathfrak{g}$ [15]

$$R([t,x]) = [t,R(x)], \tag{14}$$

or equivalently

$$(\mathrm{ad}_t \otimes 1 + 1 \otimes \mathrm{ad}_t)(r) = 0. \tag{15}$$

---

[4]Regardless of reality conditions on the $R$ operator, it is not possible to construct antisymmetric $R$ operators mixing real even and odd generators. The apparent claims to the contrary in [6] use non-real elements of the complexified algebra.

## 3  Unimodular jordanian $r$ matrices

Below we will extract several unimodular jordanian $r$ matrices from inhomogeneous $r$ matrices. Before doing so, let us introduce unimodularity, and discuss a way of finding unimodular jordanian $r$ matrices for a given superalgebra. The background of a Yang-Baxter superstring will satisfy the regular supergravity equations of motion provided the $r$ matrix satisfies the unimodularity condition [21]

$$\mathcal{K}^{mn}\mathrm{sTr}\left([t_m, R(t_n)\}x\right) = 0, \quad \forall x \in \mathfrak{g}, \tag{16}$$

where $\mathcal{K}$ with upper indices is the inverse to $\mathcal{K}_{mn} = \mathrm{sTr}(t_m t_n)$. Expressed in terms of $r$, this becomes[5]

$$r^{ij}[t_i, t_j\} = 0. \tag{17}$$

The jordanian $r$ matrix (12) is clearly not unimodular, but that can be fixed.

Let us consider $\mathcal{W}$, the $\mathcal{N} = 1$ super Weyl algebra in one dimension,

$$[d, p^0] = p^0, \quad [d, Q_i] = \tfrac{1}{2}Q_i, \quad \{Q_i, Q_j\} = -4i\delta_{ij}p^0, \tag{18}$$

where $d$ and $p^0$ are the dilatation and momentum generator respectively, and the $Q_i$, $i = 1, 2$ are two supercharges. Based on this algebra we can construct the basic jordanian $r$ matrix

$$r = d \wedge p^0. \tag{19}$$

Checking for unimodularity we find $r^{ij}[t_i, t_i\} = 2p^0$. Since this is basically $\{Q, Q\}$, we can try to make the $r$ matrix unimodular by adding $Q \wedge Q$ terms. Starting with the ansatz $r = d \wedge p^0 + aQ_1 \wedge Q_1 + bQ_2 \wedge Q_2$, unimodularity requires $b = i/4 - a$, and the classical Yang-Baxter equation subsequently fixes $a = i/8$. In other words

$$r = d \wedge p^0 + \tfrac{i}{8}(Q_1 \wedge Q_1 + Q_2 \wedge Q_2) \tag{20}$$

is a unimodular jordanian $r$ matrix. Note that the factor of $i$ makes the $R$ operator real, in line with the discussion in the previous section.

This can be repeated in higher dimensions, but is more involved since anticommutators of supercharges give combinations of momentum generators. In four dimensions, for instance,

$$\{Q_i^I, Q_j^J\} = -2i\delta^{IJ}(\sigma_\mu^s)_{ij}p^\mu, \qquad \{Q_{i+2}^I, Q_{j+2}^J\} = -2i\delta^{IJ}(\sigma_\mu^s)_{ij}p^\mu,$$
$$\{Q_i^I, Q_{j+2}^J\} = -2\delta^{IJ}(\sigma_\mu^a)_{ij}p^\mu,$$

where $i$ and $j$ take values 1 or 2, the $R$ symmetry indices $I$ and $J$ run from 1 to $\mathcal{N}$, $\sigma_0 = 1_{2\times2}$, the $\sigma_i$ are the Pauli matrices, and $\sigma^{s/a} = \sigma \pm \sigma^t$. To find a unimodular extension of

$$r = d \wedge p^\mu, \tag{21}$$

we look for a subalgebra of the form of the super Weyl algebra $\mathcal{W}$.[6] However, the square of any supercharge necessarily produces $p^0$ in addition to possible other momenta, so that we cannot cover the spacelike case in this way. For the lightlike case, we can take $Q_1$ and $Q_3$ (any $R$ symmetry indices) with $\{Q_1, Q_1\} = \{Q_3, Q_3\} = -4i(p^0 + p^3)$ and $\{Q_1, Q_3\} = 0$, to construct

$$r = d \wedge (p^0 + p^3) - \tfrac{i}{8}(Q_1 \wedge Q_1 + Q_3 \wedge Q_3), \tag{22}$$

---

[5]An equivalent definition of unimodularity is that the (super)trace of the structure constants of the Lie (super)algebra with bracket $[x, y]_R = [R(x), y] + [x, R(y)]$ is zero.

[6]We are guaranteed that one such subalgebra exists in the superconformal case, $\mathcal{W} \subset \mathfrak{psu}(1, 1|1) \subset \mathfrak{su}(2, 2|\mathcal{N})$, but this is not manifest when considering just the higher dimensional super Weyl algebra, and appears not to be true in the $\mathcal{N} = 1$ case.

which is unimodular and solves the classical Yang-Baxter equation. Within the $\mathcal{N} = 1$ super Weyl algebra in four dimensions it is not possible to make the timelike case unimodular in this fashion. If we consider $\mathcal{N} = 2$ however, we can construct

$$r = d \wedge p^0 - \tfrac{i}{16}\left(Q \wedge Q + \tilde{Q} \wedge \tilde{Q}\right), \quad Q = Q_1^1 + Q_2^2, \quad \tilde{Q} = Q_3^1 + Q_4^2. \tag{23}$$

This shows that unimodular extended jordanian $r$ matrices exist, answering the question raised in [21].[7] We will not systematically discuss further $r$ matrices of this type, but we will give further examples below. We should also note that in [21] it was claimed that unimodular extensions of in particular $d \wedge p^\mu$-type $r$ matrices do not exist – our results show that this claim is incorrect at least in the cases $\mu = 0, \pm$.[8]

## 4 Boosts, $r$ matrices, and string backgrounds

Our new unimodular jordanian $r$ matrices can be used to find new supergravity backgrounds associated to integrable string sigma models. We could extract these backgrounds by following [21]. We instead prefer to describe several unimodular jordanian models as limits of inhomogeneous deformed models, following [20]. In this picture, inhomogeneous $r$ matrices can be "boosted" to homogeneous ones, and these boosts can be implemented via a particular coordinate scaling in the background. We will generate unimodular homogeneous $r$ matrices and backgrounds by applying this idea to the unimodular $r$ matrices and backgrounds of [12].[9]

### 4.1 $\mathbf{AdS_2 \times S^2 \times T^6}$

*r* **matrix** The relevant superalgebra in this case is $\mathfrak{psu}(1,1|2)$. Working in a four by four matrix representation of $\mathfrak{su}(1,1|2)$ the standard inhomogeneous non-unimodular $r$ matrix is

$$r = -i e_j \wedge f^j, \tag{24}$$

where the $e_j$ and $f^j$ are the positive and negative roots of $\mathfrak{psu}(1,1|2)$, given by the matrices

$$\begin{pmatrix} 0 & \tilde{e}_1 & \tilde{e}_2 & \tilde{e}_3 \\ \tilde{f}^1 & 0 & \tilde{e}_4 & \tilde{e}_5 \\ \tilde{f}^2 & \tilde{f}^4 & 0 & \tilde{e}_6 \\ \tilde{f}^3 & \tilde{f}^5 & \tilde{f}^6 & 0 \end{pmatrix}, \tag{25}$$

where we take $\tilde{e}_j = 1$ for $e_j$ and the rest zero, and similarly for $f$. If we now do a noncompact group transformation $r \to \mathrm{Ad}_b \otimes \mathrm{Ad}_b (r)$ by

$$b = \begin{pmatrix} \cosh\frac{\beta}{2} & \sinh\frac{\beta}{2} & 0 & 0 \\ \sinh\frac{\beta}{2} & \cosh\frac{\beta}{2} & 0 & 0 \\ 0 & 0 & 1 & 0 \\ 0 & 0 & 0 & 1 \end{pmatrix}, \tag{26}$$

the even piece of the $r$ matrix transforms nontrivially. We then consider $\eta\, r$, with $\eta = \alpha e^{-\beta}$, and take the limit $\beta \to \infty$ to generate a new, homogeneous, $r$ matrix

$$\lim_{\beta \to \infty} \eta\, r = \alpha\, d \wedge p^0, \tag{27}$$

---

[7]Regarding the spacelike case, we might be tempted to try and subtract the unimodular version of $d \wedge (p^0 + p^3)$ from the one for $d \wedge (p^0 - p^3)$ to get a unimodular version of $d \wedge p^3$. This does not solve the classical Yang-Baxter equation however. It is possible to find unimodular extensions in the spacelike case, but this seems to require adding extra bosonic generators. We give an example of this in the conclusions.

[8]We thank Riccardo Borsato for bringing this to our attention.

[9]Section 3 notwithstanding, this is actually how we found unimodular jordanian $r$ matrices for $\mathfrak{psu}(2,2|4)$.

where $d$ and $p$ are the dilatation and momentum generator in $\mathfrak{psu}(1,1|2)$, see appendix A.1 for our conventions. The odd contributions to the $r$ matrix simply drop out, as the sum over odd roots results in a scalar with respect to $SU(1,1)$ transformations like $b$ – it is of the schematic form $r_f = \sum v_i \wedge w^i$, where $v \to bv$ and $w \to b^{-1}w$. Starting from a non-unimodular $r$ matrix, we finish with a non-unimodular $r$ matrix, and we will never generate contributions from odd generators in this way.

In [12] it was observed that doing the permutation

$$r \to r_P = \mathrm{Ad}_P \otimes \mathrm{Ad}_P\,(r), \qquad P = \begin{pmatrix} 1 & 0 & 0 & 0 \\ 0 & 0 & 1 & 0 \\ 0 & 1 & 0 & 0 \\ 0 & 0 & 0 & 1 \end{pmatrix}, \tag{28}$$

gives a unimodular inhomogeneous $r$ matrix for $\mathfrak{psu}(1,1|2)$. This permutation does not affect the even part of the $r$ matrix, and $b$ acts the same there. However, due to the permutation, the odd part of the $r$ matrix is no longer invariant, and taking the limit we find precisely the unimodular jordanian $r$ matrix discussed above,

$$\lim_{\beta \to \infty} \eta\, r_P = \alpha\left(d \wedge p^0 + \tfrac{i}{8}(Q_1^1 \wedge Q_1^1 + Q_2^2 \wedge Q_2^2)\right), \tag{29}$$

with the $Q_i$ given in appendix A.1.

As an aside, regardless of unimodularity, finding new solutions to the classical Yang-Baxter equation is not simple, and a brute force approach quickly becomes prohibitive for larger algebras. The above infinite boost story offers an efficient, albeit inherently limited, way to generate new solutions based on an inhomogeneous $r$ matrix. From this perspective, by breaking the invariance of the odd contributions to the standard inhomogeneous $r$ matrix with respect to bosonic group transformations, the permutation allows us to find new $r$ matrices with contributions from odd generators.

**Supergravity background**  The supergravity background of the Yang-Baxter sigma model for the original $r_P$ is [12]

$$
\begin{aligned}
ds^2 &= \frac{1}{1-\kappa^2\rho^2}\left(-(1+\rho^2)dt^2 + \frac{d\rho^2}{1+\rho^2}\right) + \frac{1}{1+\kappa^2 r^2}\left((1-r^2)d\phi^2 + \frac{dr^2}{1-r^2}\right) + dx^i dx^i \,, \\
B &= -\frac{\kappa\rho}{1-\kappa^2\rho^2}dt \wedge d\rho - \frac{\kappa r}{1+\kappa^2 r^2}d\phi \wedge dr \,, \\
\mathcal{F}_3 &= -N\left(\kappa r(1+\kappa^2 r^2)d\rho + \kappa\rho(1-\kappa^2\rho^2)dr\right)\wedge J_2 \\
&\quad + N\left(\kappa\rho(1-r^2)d\rho - \kappa r(1+\rho^2)dr\right)\wedge dt \wedge d\phi \,, \\
\mathcal{F}_5 &= N\left((1+\kappa^2 r^2)d\rho - \kappa^2\rho r(1+\rho^2)dr\right)\wedge dt \wedge \mathrm{Re}\,\Omega_3 \\
&\quad - N\left(\kappa^2\rho r(1-r^2)d\rho + (1-\kappa^2\rho^2)dr\right)\wedge d\phi \wedge \mathrm{Im}\,\Omega_3 \,, \\
e^{-2\Phi} &= e^{-2\Phi_0}\frac{(1-\kappa^2\rho^2)(1+\kappa^2 r^2)}{1-\kappa^2(\rho^2 - r^2 - \rho^2 r^2)} \,,
\end{aligned}
\tag{30}
$$

where the $x_i$, $i = 4, \ldots, 9$ are flat coordinates on $T^6$, and

$$
\begin{aligned}
N &= \frac{\sqrt{1+\kappa^2}}{\sqrt{1-\kappa^2\rho^2}\sqrt{1+\kappa^2 r^2}}\frac{1}{1-\kappa^2(\rho^2 - r^2 - \rho^2 r^2)} \,, \\
\Omega_3 &= dz^1 \wedge dz^2 \wedge dz^3 \,, \\
J_2 &= \frac{i}{2}(d\bar{z}^1 \wedge dz^1 + d\bar{z}^2 \wedge dz^2 + d\bar{z}^3 \wedge dz^3),
\end{aligned}
\tag{31}
$$

with $z^1 = x^4 - ix^5$, $z^2 = x^6 - ix^7$, and $z^3 = x^8 - ix^9$. The $\mathcal{F}_i$ are related to the standard RR forms $F_i$ as $\mathcal{F}_i = e^\Phi F_i$. This background coincides with the one parameter family of backgrounds of [32] evaluated at their $a = 1$.

Boosting an $r$ matrix by $b$ and taking $\beta \to \infty$ is equivalent to rescaling

$$t \to 2e^{-\beta}t, \qquad \rho \to e^\beta \frac{1}{2z}, \tag{32}$$

and taking $\beta \to \infty$ in the Yang-Baxter sigma model background, as explained in [20]. Doing so in the above gives

$$
\begin{aligned}
ds^2 &= \frac{-dt^2 + dz^2}{z^2 - \alpha^2} + (1 - r^2)d\phi^2 + \frac{dr^2}{1 - r^2} + dx^i dx^i\,, \\
B &= -\frac{\alpha}{z}\frac{dt \wedge dz}{z^2 - \alpha^2}, \\
F_3 &= \frac{\alpha e^{-\Phi_0}}{z^2 - \alpha^2(1 - r^2)}\big((rz\,dz - (z^2 - \alpha^2)dr) \wedge J_2 - ((1 - r^2)dz + rz\,dr) \wedge dt \wedge d\phi\big), \\
F_5 &= \frac{e^{-\Phi_0}}{z^2 - \alpha^2(1 - r^2)}\big((z\,dt \wedge dz + \alpha^2 r\,dt \wedge dr) \wedge \text{Re}\,\Omega_3 \\
&\quad + (z(z^2 - \alpha^2)d\phi \wedge dr + \alpha^2 r(1 - r^2)dz \wedge d\phi) \wedge \text{Im}\,\Omega_3\big), \\
e^{-2\Phi} &= e^{-2\Phi_0}\frac{z^2 - \alpha^2}{z^2 - \alpha^2(1 - r^2)}.
\end{aligned}
\tag{33}
$$

This is our first example of a jordanian supergravity background, and the first example of a background associated to a homogeneous deformation involving odd generators.

The deformation parameter can be completely absorbed, by rescaling $t \to \alpha t$, $z \to \alpha z$, and shifting $\Phi_0 \to \Phi_0 + \alpha$. This is in line with the discussion in [33], as our $r$ matrix corresponds to a co-boundary cocycle $\omega$ and hence the deformation should reduce to just a non-abelian T duality transformation, with no intrinsic deformation parameter.[10] Interestingly, we can also scale out the deformation parameter from the $r$ matrix in an a priori different way. Under the automorphism of the superconformal algebra given by

$$p \to p/\alpha, \quad k \to \alpha k, \quad Q \to Q/\sqrt{\alpha}, \quad S \to \sqrt{\alpha}S, \tag{34}$$

where our $r$ matrix transforms as $\alpha r \to r$. From this perspective the deformation parameter can hence be changed by an algebra automorphism, which does affect the model. It would be interesting to understand whether these automorphism and co-boundary discussions are related. Similar comments apply to all $r$ matrices and backgrounds given below, but we have not explicitly checked that all $r$ matrices correspond to co-boundary cocycles.

Finally, it is interesting to note that this deformation of $\text{AdS}_2 \times \text{S}^2 \times \text{T}^6$ gives back undeformed $\text{AdS}_2 \times \text{S}^2 \times \text{T}^6$ upon timelike T duality in $t$, albeit supported by non-standard fluxes and a nontrivial dilaton, as a type II* supergravity solution [34].

## 4.2 $\text{AdS}_5 \times \text{S}^5$

The transformation $b$ we used above for $\text{AdS}_2$ is just a dilation $b = e^{\beta d}$, which up to rotations is the only noncompact transformation available there. For $\text{AdS}_5$ we have multiple inequivalent choices available to us. We will start with an infinite dilation, as before. Although the matrices and backgrounds have larger expressions in this case, the procedure is exactly the same.

---

[10]Restricted to $\tilde{\mathfrak{g}} = \text{span}(\{d, p^0, Q_1^1, Q_2^2\})$, the $R$ operator has a right inverse $\Omega$, in our conventions associated to $\omega = 2d \wedge k^0 + \frac{i}{8}(S_1^1 \wedge S_1^1 + S_2^2 \wedge S_2^2))$ like $R$ is associated to $r$. ($\Omega$ is a left inverse when restricted to $\text{span}(\{d, k^0, S_1^1, S_2^2\})$.) This inverse is co-boundary in the sense that $\omega$ viewed as a map from $\tilde{\mathfrak{g}} \otimes \tilde{\mathfrak{g}} \to \mathbb{R}$ can be written in the form $\omega(x, y) = f([x, y])$ for some linear function $f : \tilde{\mathfrak{g}} \to \mathbb{R}$. In the present case $f(x) = -s\text{Tr}(k^0 x)$. We thank Riccardo Borsato and Ben Hoare for related discussions.

### 4.2.1 Infinite dilation

*r* **matrix**   Starting from the permuted $R$ operator given in equation (4.7) of [12], for the associated $r$ matrix $r_P$ we find

$$\lim_{\beta \to \infty} (\mathrm{Ad}_b \otimes \mathrm{Ad}_b)(\eta\, r_P) = \tfrac{\alpha}{2}(r_0 + r_1), \tag{35}$$

with

$$
\begin{aligned}
r_0 &= d \wedge p^0 + M^0{}_\mu \wedge p^\mu - M^1{}_2 \wedge p^2 - M^1{}_3 \wedge p^3, \\
r_1 &= -\tfrac{i}{8}\sum_{j=1}^{4} Q_j^2 \wedge Q_j^2 - \tfrac{i}{16}\sum_{I=1,3}(Q_1^I - Q_2^I)\wedge(Q_1^I - Q_2^I) + (Q_3^I - Q_4^I)\wedge(Q_3^I - Q_4^I).
\end{aligned}
\tag{36}
$$

Our conventions for $\mathfrak{psu}(2,2|4)$ can be found in appendix A.2.

**Supergravity background**   To implement this limit in the supergravity background of section 4.2 of [12], we rescale and relabel the coordinates used there as [20]

$$t \to 2e^{-\beta} t, \quad \rho \to e^\beta \frac{1}{2z}, \quad x \to 2e^{-\beta}\rho, \quad \psi_1 \to -2e^{-\beta}x, \quad \psi_2 \to \theta. \tag{37}$$

In the limit $\beta \to \infty$ with $\eta = e^{-\beta}\alpha$, we then find the NSNS background fields

$$
\begin{aligned}
ds^2 &= \frac{-dt^2 + dz^2}{z^2 - \alpha^2} + \frac{d\rho^2 + dx^2}{z^2 + \alpha^2 z^{-2}\rho^2} + \frac{\rho^2 d\theta^2}{z^2} \\
&\quad + (1-r^2)d\phi^2 + \frac{dr^2}{1-r^2} + \frac{r^2 dw^2}{1-w^2} + r^2(1-w^2)d\phi_1^2 + r^2 w^2 d\phi_2^2, \\
B &= -\frac{\alpha}{z}\frac{dt \wedge dz}{z^2 - \alpha^2} + \frac{\alpha\rho\, d\rho \wedge dx}{z^4 + \rho^2\alpha^2}, \\
e^{-2\Phi} &= e^{-2\Phi_0}\frac{z^2 - \alpha^2}{z^2}\frac{f_1}{f_2^2},
\end{aligned}
\tag{38}
$$

where

$$f_1 = z^2 + \alpha^2 y^2, \quad f_2 = z^2 - \alpha^2(1 - r^2(1-w^2)). \tag{39}$$

The RR fields can be expressed as

$$F_3 = dC_2, \qquad F_5 = dC_4 + H \wedge C_2 = (1 + \star)(dC_{4|t} + H \wedge C_2), \tag{40}$$

where $C_{4|t}$ denotes the part of $C_4$ proportional to $dt$, and

$$
\begin{aligned}
e^{\Phi_0}C_2 &= \frac{\alpha}{f_2}\left((1-r^2)dt \wedge d\phi + r^2 w^2 dt \wedge d\phi_2 + \frac{\rho^2}{z^2}dt \wedge d\theta + r^2(1-w^2)dx \wedge d\phi_1\right), \\
e^{\Phi_0}C_{4|t} &= \frac{(rz\, dt \wedge dx \wedge dr + (1-r^2)dt \wedge dz \wedge dx)\wedge(d\phi_2 - d\phi)}{z f_2} \\
&\quad + \frac{\rho\, dt \wedge d\rho \wedge dx \wedge(z^2(d\phi_2 - d\theta) - \alpha^2(1-r^2)(d\phi_2 - d\phi))}{f_1 f_2} \\
&\quad - \frac{2z^3 dt \wedge dz \wedge dx \wedge d\phi_2}{\alpha^2 f_1}.
\end{aligned}
\tag{41}
$$

The $\alpha \to 0$ divergence in the last term of $C_{4|t}$ is a gauge artifact, $F_5$ has a well-defined undeformed limit. $\alpha$ can be absorbed by rescaling $t, z, \rho$, and $x$, and shifting the dilaton.

#### 4.2.2 Alternate infinite boost

As discussed in [20], at the bosonic level, up to finite group transformations there is one other inequivalent boost we can use on our inhomogeneous $r$ matrix. In our conventions this boost is generated by $p^2 + k^2$, and results in the (unimodular) abelian $r$ matrix $r = p^1 \wedge p^2$. The permutation of the original inhomogeneous $r$ matrix does not affect this result.

#### 4.2.3 Subsequent infinite Lorentz boost

In addition to inequivalent initial transformations, we can also perform them sequentially and potentially get something new. In particular, we can Lorentz boost the deformation we found above.

**$r$ matrix**  With $\alpha = \sqrt{2}e^{-\beta}\gamma$ and $\tilde{b} = e^{-\beta M^{01}}$, boosting the $r$ matrix (35) gives

$$\lim_{\beta \to \infty} \left( \text{Ad}_{\tilde{b}} \otimes \text{Ad}_{\tilde{b}} \right) (\alpha(r_0 + r_1)) = \gamma(r_0^+ + r_1^+), \tag{42}$$

where

$$
\begin{aligned}
r_0^+ &= (d + M_+^+) \wedge p^+, \\
r_1^+ &= -\tfrac{i}{8} \left( (Q_1^2 + Q_2^2) \wedge (Q_1^2 + Q_2^2) + (Q_3^2 + Q_4^2) \wedge (Q_3^2 + Q_4^2) \right),
\end{aligned}
\tag{43}
$$

and we have introduced light-cone coordinates as $\sqrt{2}v^{\pm} = v^0 \pm v^1$. This $r$ matrix is precisely of the schematic form of the $r$ matrix (22) – the underlying algebra is identical. We can also boost oppositely, but we focus on this case as the resulting $r$ matrix is simpler.

**Supergravity background**  To implement our infinite Lorentz boost on the supergravity background of equations (38,41), we just rescale the light-cone coordinates $x^{\pm} = (t \pm x)/\sqrt{2}$ as $x^{\pm} \to e^{\mp\beta}$, take $\alpha = \sqrt{2}e^{-\beta}\gamma$, and $\beta \to \infty$. This gives

$$
\begin{aligned}
ds^2 &= -\frac{2dx^+ dx^- + dz^2 + d\rho^2}{z^2} + \frac{\rho^2}{z^2}d\theta^2 - \gamma^2 \frac{z^2 + \rho^2}{z^6}(dx^-)^2 + d\Omega_5^2, \\
B &= \frac{\gamma}{z^4}(zdz - \rho d\rho) \wedge dx^-, \\
F_3 &= \frac{2\gamma e^{-\Phi 0}}{z^3} \big[ (1 - r^2)dz \wedge d\phi - r^2(1 - w^2)dz \wedge d\phi_1 + w^2 dz \wedge d\phi_2 \\
&\quad - rz(d\phi \wedge dr + (1 - w^2)d\phi_1 \wedge dr - w^2 d\phi_2 \wedge dr) \\
&\quad - r^2 wz dw \wedge (d\phi_1 + d\phi_2) - \frac{\rho}{z}d\rho \wedge d\theta + \frac{2\rho}{z^2}dz \wedge d\theta \big] \wedge dx^-, \\
F_5 &= F_5^0, \quad \Phi = \Phi_0,
\end{aligned}
\tag{44}
$$

where $F_5^0$ is the undeformed $AdS_5 \times S^5$ five form. $\gamma$ can again be absorbed.

The NSNS background of this model by construction is the same as the one associated to the purely even $r$ matrix $r_0^+$. Although not written in exactly this form, this $r$ matrix appeared before in [6], and a supergravity background supporting the associated NSNS background was proposed in [30]. However, the RR fluxes of that background break the SO(6) invariance associated to the sphere, while the $r$ matrix does not, and so the two are incompatible, in addition to the now understood situation regarding unimodularity and supergravity. Interestingly, due to $r_1^+$ our $r$ matrix is not SO(6) invariant, but it is unimodular (supergravity). These facts all nicely match up, and indeed it turns out that our background is identical to the one of [30].[11]

---

[11]This is manifest after the diffeomorphism $r = \sin(\mu), w = \sin(\tilde{\theta}/2), \phi = \chi, \phi_1 = -(2\chi + \psi + \tilde{\phi})/2$, $\phi_2 = -(2\chi + \psi - \tilde{\phi})/2$, up to some tildes to distinguish from our labeling of AdS variables.

This same background was also used as an example in [8], where a noncommutative dual field theory interpretation of homogeneous Yang-Baxter deformed strings was proposed. According to the general conjecture of [8], while the bosonic analysis of this example is unmodified, our extension by $r_1^+$ induces further fermionic noncommutativity in the dual field theory.

Interestingly, this background can also be obtained from undeformed AdS$_5\times$S$^5$ by two TsT transformations and an S-duality transformation [35]. Since homogeneous deformations are equivalent to non-abelian T-duality transformations [33, 36], this shows that, as a solution generating technique, non-abelian T duality with respect to a superalgebra can be equivalent to a U-duality transformation that involves S duality. Moreover, we note that since Yang-Baxter deformations preserve integrability, this manifests integrability of the superstring in this S-dual background.

## 5  Conclusions

In this paper we found jordanian $r$ matrices that are unimodular, and extracted examples of backgrounds for the associated Yang-Baxter superstring sigma models. These results are based on the unimodular inhomogeneous $r$ matrices of [12], which from the boosting perspective have the special property that their fermionic terms are not invariant with respect to bosonic group transformations. We have not tried to give an exhaustive overview of the possible $r$ matrices that can be obtained this way, and there are further inequivalent boosts and permutations that we could consider. For instance, for $\mathfrak{psu}(2,2|4)$, by combining $P$ from [12] with the "permutation" $P_2$ from [11] to form $r_{PP_2}$ as a starting point, we can find the unimodular $r$ matrix

$$r = d \wedge p^1 + M^1{}_0 \wedge p^0 + \tfrac{i}{16}(q_1 \wedge q_1 + q_2 \wedge q_2 - q_3 \wedge q_3 - q_4 \wedge q_4),$$
$$q_1 = Q_1^1 - Q_2^1, \quad q_2 = Q_3^1 - Q_4^1, \quad q_3 = Q_1^3 + Q_2^3, \quad q_4 = Q_3^3 + Q_4^3.$$

This $r$ matrix is of a form that appears to fall solidly outside the scope of the discussion in section 3, given the basic $d \wedge p^1$ term.[12] It would be interesting to understand the full space of inequivalent homogeneous $r$ matrices, purely bosonic or not, unimodular or not, that can be obtained by external automorphisms (permutations) and infinite boosts.

Beyond this, as discussed above, our last example shows that non-abelian T duality with respect to a superalgebra can give the same background as an S-duality transformation, up to some further T dualities. It would be interesting to see whether this is an accident in this specific case, or indicative of something systematic. Since homogeneous Yang-Baxter deformations, or equivalently non-abelian T duality, preserve integrability, this would manifest integrability for any such S-dual background, via explicit Lax pairs.

Next, at the algebraic level, homogeneous deformations are determined by Drinfeld twists. These Drinfeld twists are in one to one correspondence with the associated $r$ matrices, but not known in general. It would be interesting to construct the twists corresponding to our new $r$ matrices.

Moreover, the types of limits we are considering here can be combined with contraction limits [37, 38] to yield deformations of flat space. It would be interesting to see if these yield interesting new models or make contact with known ones.

Finally, when it comes to these and other deformations of the superstring, with the exception of particular diagonal abelian deformations [39, 40] and inhomogeneous deformations [41, 42], efficiently describing them at the quantum level appears to be a complicated open problem, see e.g. the discussion in [43]. It might be interesting to first study them in

---

[12]Note that $2(d \wedge p^1 + M^1{}_0 \wedge p^0) = (d + M^1{}_0) \wedge (p^0 + p^1) - (d - M^1{}_0) \wedge (p^0 - p^1)$, but whether this has more meaning than $2d \wedge p^1 = d \wedge (p^0 + p^1) - d \wedge (p^0 - p^1)$ is not immediately clear.

simplified setups such as the plane wave limit.[13]

## Acknowledgments

I would like to thank Riccardo Borsato, Ben Hoare and Linus Wulff for helpful discussions and comments on the draft. The work of the author is supported by the German Research Foundation (DFG) via the Emmy Noether program "Exact Results in Extended Holography". I am supported by L.T.

## A  $\mathfrak{psu}(1,1|2)$ and $\mathfrak{psu}(2,2|4)$

It is convenient to think of the superalgebras $\mathfrak{psu}(1,1|2)$ and $\mathfrak{psu}(2,2|4)$ in terms of the $4\times4$ and $8\times8$ supermatrix realizations of $\mathfrak{su}(1,1|2)$ and $\mathfrak{su}(2,2|4)$, modulo the ideals generated by the identity matrices. As a general definition we take $\mathfrak{su}(1,1|2)$ as the space of $4\times4$ supermatrices that satisfy the reality condition

$$M^\dagger H + HM = 0, \quad H = \text{diag}(1,-1,1,1), \tag{45}$$

while for $\mathfrak{su}(2,2|4)$ we have $8\times8$ matrices and $H = \text{diag}(1,1,-1,-1,1,1,1,1)$. Below we discuss our choice of basis and the associated commutation relations in each case.

### A.1  $\mathfrak{psu}(1,1|2)$

For $\mathfrak{su}(1,1|2)$ we can work in the matrix conventions of [45], identifying their generators $J_{01}$, $P_0$, $P_1$ and $Q_{I\hat{\alpha}\check{\alpha}}$ as

$$
\begin{aligned}
p^0 &= -(P_0 + P_1), & k^0 &= -(P_0 - P_1), & d &= -J_{01}, \\
Q_1^1 &= Q_{111} + Q_{121}, & Q_1^2 &= Q_{112} + Q_{122}, \\
Q_2^2 &= Q_{211} + Q_{221}, & Q_2^1 &= Q_{212} + Q_{222}, \\
S_1^1 &= Q_{211} - Q_{221}, & S_1^2 &= Q_{212} - Q_{222}, \\
S_2^2 &= Q_{121} - Q_{111}, & S_2^1 &= Q_{122} - Q_{112}.
\end{aligned}
\tag{46}
$$

Here $d$, $p^0$ and $k^0$ denote respectively the dilatation, momentum and special conformal generators, and $Q_i$ and $S_i$ the (real) regular and conformal supercharges. Together with three generators for the $\mathfrak{su}(2)$ $R$ symmetry algebra, they form the one dimensional $\mathcal{N}=2$ superconformal algebra

$$
\begin{aligned}
[d, p^0] &= p^0, & [d, k^0] &= -k^0, & [p^0, k^0] &= -2d, \\
[d, Q_i] &= \tfrac{1}{2}Q_i, & [d, S_i] &= -\tfrac{1}{2}S_i, \\
\{Q_i^I, Q_j^J\} &= -4i\delta^{IJ}\delta_{ij}p^0, & \{S_i^I, S_j^J\} &= -4i\delta^{IJ}\delta_{ij}k^0, & \{Q,S\} &= 0,
\end{aligned}
\tag{47}
$$

where $(Q^1, Q^2)$ and $(S^1, S^2)$ transform in the fundamental representation of the $\mathfrak{su}(2)$ $R$ symmetry.

---

[13]See e.g. [44] for the plane wave limit of the inhomogeneous deformation of $\text{AdS}_5 \times S^5$.

## A.2  $\mathfrak{psu}(2,2|4)$

For $\mathfrak{so}(2,4) \simeq \mathfrak{su}(2,2) \subset \mathfrak{su}(2,2|4)$ we work in the conventions of [46], defining

$$m^{ij} = \frac{1}{4}[\gamma^i, \gamma^j], \qquad m^{i5} = -m^{5i} = \frac{1}{2}\gamma^i, \qquad i = 0, \dots, 4, \tag{48}$$

where

$$\begin{aligned}
\gamma^0 &= i\sigma_3 \otimes \sigma_0, & \gamma^1 &= \sigma_2 \otimes \sigma_2, & \gamma^2 &= -\sigma_2 \otimes \sigma_1, \\
\gamma^3 &= \sigma_1 \otimes \sigma_0, & \gamma^4 &= \sigma_2 \otimes \sigma_3, & \gamma^5 &= -i\gamma^0,
\end{aligned} \tag{49}$$

$\sigma_0 = \mathbf{1}_{2 \times 2}$ and $\sigma_a$ are the Pauli matrices. These $m^{ij}$ satisfy the standard $\mathfrak{so}(2,4)$ commutation relations

$$[m^{ij}, m^{kl}] = \eta^{jk} m^{il} - \eta^{ik} m^{jl} - \eta^{jl} m^{ik} + \eta^{il} m^{jk}, \qquad i,j,k,l = 0, \dots, 5, \tag{50}$$

where $\eta = \mathrm{diag}(-1,1,1,1,1,-1)$. We define the corresponding conformal generators as in [20]

$$\begin{aligned}
p^\mu &= m^{\hat{\mu}0} - m^{\hat{\mu}1}, & M^{\mu\nu} &= m^{\hat{\mu}\hat{\nu}}, \\
k^\mu &= m^{\hat{\mu}0} + m^{\hat{\mu}1}, & d &= -m^{01},
\end{aligned} \tag{51}$$

where[14]

$$\hat{\mu} = \begin{cases} 5 & \mu = 0, \\ i+1 & \mu = i \in \{1,2,3\}. \end{cases} \tag{52}$$

We then construct our supercharges $Q_j^I$ as follows. We start with

$$(\mathcal{Q}_1)_{ij} = 2\delta_{i1}\delta_{j5}, \quad (\mathcal{Q}_2)_{ij} = 2\delta_{i2}\delta_{j5}, \tag{53}$$

and $\bar{\mathcal{Q}}$ such that $\mathcal{Q} + \bar{\mathcal{Q}}$ is real. We then take

$$\begin{aligned}
Q_1^1 &= \mathrm{Ad}_M(\mathcal{Q}_1 + \bar{\mathcal{Q}}_1), & Q_2^1 &= \mathrm{Ad}_M(\mathcal{Q}_2 + \bar{\mathcal{Q}}_2), \\
Q_3^1 &= i\mathrm{Ad}_M(\mathcal{Q}_1 - \bar{\mathcal{Q}}_1), & Q_4^1 &= i\mathrm{Ad}_M(\mathcal{Q}_2 - \bar{\mathcal{Q}}_2),
\end{aligned} \tag{54}$$

where[15]

$$M = \mathrm{diag}(T, \mathbb{I}), \quad T = \frac{1}{2\sqrt{2}} \begin{pmatrix} 1+i & -1-i & -1+i & -1+i \\ -1-i & -1-i & -1+i & 1-i \\ 1+i & -1-i & 1-i & 1-i \\ -1-i & -1-i & 1-i & -1+i \end{pmatrix}. \tag{55}$$

$Q_i^1$ has only nonzero entries in the fifth row and column, the $Q_i^I$ have the same entries, just in row and column $I+4$. The $S_j^I$ are constructed similarly, starting from $(\mathcal{S}_1)_{ij} = -2\delta_{i5}\delta_{j2}$, and $(\mathcal{S}_2)_{ij} = 2\delta_{i5}\delta_{j1}$.

---

[14]This choice differs from the one in [20] by the permutation of Lorentz type $\mu$ indices $1 \leftrightarrow 2$, to match the $\sigma_i p^i$ terms in the superconformal algebra.

[15]The inverse of this transformation puts all our superconformal generators in a standard block form, with e.g. the momenta sitting in the upper right $2 \times 2$ block of the upper left $4 \times 4$ block of $\mathfrak{su}(2,2|4)$.

Together with the $\mathfrak{su}(4)$ $R$ symmetry generators, these generators satisfy the $\mathcal{N} = 4$ superconformal algebra

$$
\begin{aligned}
[M^{\mu\nu}, p^\rho] &= \eta^{\nu\rho} p^\mu - \eta^{\mu\rho} p^\nu, && [M^{\mu\nu}, k^\rho] = \eta^{\nu\rho} k^\mu - \eta^{\mu\rho} k^\nu, \\
[M^{\mu\nu}, d] &= 0, && [d, p^\mu] = p^\mu, && [d, k^\mu] = -k^\mu, \\
[p^\mu, k^\nu] &= 2M^{\mu\nu} + 2\eta^{\mu\nu} d, && [M^{\mu\nu}, M^{\rho\sigma}] = \eta^{\mu\rho} M^{\nu\sigma} + \text{perms.}, \\
[d, Q] &= \tfrac{1}{2} Q, \quad [d, S] = -\tfrac{1}{2} S, \\
\{Q_i^I, Q_j^J\} &= -2i\delta^{IJ} (\sigma_\mu^s)_{ij} p^\mu, && \{Q_{i+2}^I, Q_{j+2}^J\} = -2i\delta^{IJ} (\sigma_\mu^s)_{ij} p^\mu, \\
\{Q_i^I, Q_{j+2}^J\} &= -2\delta^{IJ} (\sigma_\mu^a)_{ij} p^\mu, \\
\{S_i^I, S_j^J\} &= -2i\delta^{IJ} (\sigma_\mu^s)_{ij} k^\mu, && \{S_{i+2}^I, S_{j+2}^J\} = -2i\delta^{IJ} (\sigma_\mu^s)_{ij} k^\mu, \\
\{S_i^I, S_{j+2}^J\} &= -2\delta^{IJ} (\sigma_\mu^a)_{ij} k^\mu,
\end{aligned}
\tag{56}
$$

where $i$ and $j$ take values 1 or 2, the $R$ symmetry indices $I$ and $J$ run from 1 to $\mathcal{N}$, $\sigma_0 = 1_{2\times 2}$, the $\sigma_i$ are the Pauli matrices, and $\sigma^{s/a} = \sigma \pm \sigma^t$. Beyond the above relations, $(Q^1, Q^2, Q^3, Q^4)$ and $(S^1, S^2, S^3, S^4)$ transform in the fundamental representation of $\mathfrak{su}(4)$. Moreover, we leave $\{Q, S\}$ and the Lorentz transformations of the $Q$ and $S$ implicitly determined by our specified matrix basis – $\text{Ad}_M(\mathcal{Q})$ and $\text{Ad}_M(\bar{\mathcal{Q}})$ transform as the standard Weyl spinors of the Lorentz group, if we take $\tilde{\gamma}_0 = \sigma_1 \otimes \sigma_0$, $\tilde{\gamma}_i = i\sigma_2 \otimes \sigma_i$.

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
