# Peer review of "Unimodular jordanian deformations of integrable superstrings"

_SciPost Physics, doi:SciPost Phys. 7, 011 (2019)_

## Round 2 · Referee Report · Riccardo Borsato (Referee 1) · 2019-6-21

Strengths

1- A class of integrable deformations of superstrings (belonging to the family of Yang-Baxter deformations) is identified. Importantly, this class had been missed in previous investigations.

2- A supergravity background (constructed in reference [34] and related to an S-duality transformation) is placed on firm ground by showing that it belongs to this new class of models. This fact also proves the (classical) integrability of the corresponding sigma-model.

3- The results are solid, and obtained by careful and trustworthy calculations.

4- The author writes his explanations in a very clear way. All details needed to reproduce the results are provided.

Weaknesses

1- A full classification of the deformations belonging to this class is missing. The task is equivalent to the classification of all superalgebras $\mathfrak{h}\subset\mathfrak{g}$, where $\mathfrak{h}$ is a unimodular superalgebra whose even part is a jordanian Lie algebra, and $\mathfrak{g}$ is either $\mathfrak{psu}(1,1|2)$ or $\mathfrak{psu}(2,2|4)$.

2- The models constructed in the paper are not strictly-speaking deformations -- when non-zero, the deformation parameter can be reabsorbed by a field redefinition. The backgrounds can be obtained by doing non-abelian T-duality on a superalgebra that is a subalgebra of the full algebra of isometries.

Report

The author obtains supergravity backgrounds by implementing some scaling limits on the backgrounds derived by Hoare and Seibold in reference [12]. This procedure is equivalent to constructing Yang-Baxter deformations of the $AdS_2\times S^2\times T^6$ and of the $AdS_5\times S^5$ superstring. These deformations are based on $R$-matrices that are solutions of the (homogeneous) classical Yang-Baxter equation, and are known to preserve the (classical) integrability of the original sigma-model.

The deformations considered in this paper are of jordanian type. As opposed to other jordanian deformations appearing previously in the literature, the ones constructed here are "unimodular" and give rise to solutions of (standard) supergravity. Unimodularity is achieved by allowing also contributions of odd elements of the superalgebra in the $R$-matrices. This paper clarifies the status of jordanian Yang-Baxter deformations, and clearly shows that some of them can have a string theory interpretation. This is an important point for the classification of superstring backgrounds that can be obtained by integrable deformations.

One may judge a weakness of the paper the fact that the supergravity backgrounds were not obtained directly from the deformation procedure (using the R-matrix as seed ingredient), rather by certain scaling limits ("singular boosts" in the language of the paper) of a "parent" supergravity background. But that would be unfair because -- while the first method is more general, and using the second method some deformations may be missed -- the two methods are perfectly equivalent for all backgrounds that can be obtained by the singular boosts.

These models deserve to be further studied. At least one of them (as argued in reference [34], where it was originally constructed) is known to be related to an S-duality transformation. These new results prove that it is in fact an integrable model. While currently the results of the paper are of interest mainly for the community of people working on integrable deformations of superstrings, they have the potential of becoming relevant for a larger community of string theorists.

The paper is very well written. The subject is quite technical, but explanations are clear and all the necessary details are provided. The list of references is also appropriate, and the introduction gives a good overview of the topic of integrable deformations of the superstring that may be useful for readers not familiar with the subject.

The paper is of high quality, and may be published already as it is. I have only minor suggestions for improvement.

Requested changes

At the end of section 4.1 it is said that undeformed $AdS_2\times S^2\times T^6$ is obtained from the previous background after doing T duality in $t$. It would be good to remind the reader that timelike T duality in fact gives a solution of type II*, rather than type II. The relevant reference is DOI: 10.1088/1126-6708/1998/07/021

There is no further important change to request. Since I noticed them, I list few typos. - In equation (3.1) the argument of R should be $t_n$ (as opposed to just $n$) to keep the notation uniform. - In footnote 10 "has an right inverse" should be "has a right inverse". - After (4.18), "an gauge artifact" should be "a gauge artifact". - After (4.20), the verb is missing in the senctence "This $r$ matrix [is] precisely [...]".

---

## Round 2 · Referee Report · Anonymous (Referee 2) · 2019-6-24

Report

This paper deals with so-called Yang-Baxter deformations of supercoset sigma models describing for example the string in AdS5xS5. New examples of homogeneous YB deformations are found by taking scaling limits of recently constructed inhomogeneous examples. In this way deformations corresponding to a (extended) jordanian R-matrix are found which satisfy the unimodularity condition, i.e. lead to genuine supergravity backgrounds, clarifying some questions in the literature.

The paper is well written and the results are of interest in the program of understanding the space of integrable deformations of superstrings. I recommend the paper for publication.

---

## Round 3 · Author Response

I thank the referees for their positive and helpful reports, in particular Riccardo Borsato for his detailed comments.

---

## Round 3 · List of Changes

With regard to the suggestions by Riccardo Borsato:
- At the end of section 4 I now mention the timelike nature of the T duality, with an appropriate reference.
- I corrected the indicated typographical errors.

Beyond the referees' suggestions:
- I made some minor changes in phrasing and indentation at various place in the paper, without affecting meaning.
- I corrected a typographical error in formula (29) and related typos in footnote 10 and equation (46).
- I added what is now footnote 12, adding a minor comment regarding the structure of the extra r matrix discussed in the discussion.

---

## Editorial Decision

published